# Further Insights on RNA Expression and Sperm Motility

**DOI:** 10.3390/genes13071291

**Published:** 2022-07-21

**Authors:** Carolina Silva, Paulo Viana, Alberto Barros, Rosália Sá, Mário Sousa, Rute Pereira

**Affiliations:** 1Laboratory of Cell Biology, Department of Microscopy, ICBAS-School of Medicine and Biomedical Sciences, University of Porto, UMIB-Unit for Multidisciplinary Research in Biomedicine, ICBAS/ITR-Laboratory for Integrative and Translational Research in Population Health, 4050-313 Porto, Portugal; carolinamsilva@outlook.pt (C.S.); rmsa@icbas.up.pt (R.S.); msousa@icbas.up.pt (M.S.); 2Faculty of Medicine, University of Coimbra (FMUC), 3000-370 Coimbra, Portugal; 3Centre for Reproductive Genetics A. Barros, 4100-012 Porto, Portugal; paulo.viana0@gmail.com (P.V.); abarros@med.up.pt (A.B.); 4Department of Genetics, Faculty of Medicine, University of Porto (FMUP), 4200-319 Porto, Portugal; 5Institute of Health Research and Innovation (IPATIMUP/i3S), University of Porto, 4200-135 Porto, Portugal

**Keywords:** gene expression, male reproduction, quantitative PCR, RNA, sperm motility

## Abstract

Asthenozoospermia is one of the main causes of male infertility and it is characterized by reduced sperm motility. Several mutations in genes that code for structural or functional constituents of the sperm have already been identified as known causes of asthenozoospermia. In contrast, the role of sperm RNA in regulating sperm motility is still not fully understood. Consequently, here we aim to contribute to the knowledge regarding the expression of sperm RNA, and ultimately, to provide further insights into its relationship with sperm motility. We investigated the expression of a group of mRNAs by using real-time PCR (*CATSPER3*, *CFAP44*, *CRHR1*, *HIP1*, *IQCG KRT34*, *LRRC6*, *QRICH2*, *RSPH6A*, *SPATA33* and *TEKT2*) and the highest score corresponding to the target miRNA for each mRNA in asthenozoospermic and normozoospermic individuals. We observed a reduced expression of all mRNAs and miRNAs in asthenozoospermic patients compared to controls, with a more accentuated reduction in patients with progressive sperm motility lower than 15%. Our work provides further insights regarding the role of RNA in regulating sperm motility. Further studies are required to determine how these genes and their corresponding miRNA act regarding sperm motility, particularly *KRT34* and *CRHR1*, which have not previously been seen to play a significant role in regulating sperm motility.

## 1. Introduction

Infertility is a disease of the reproductive system that can affect both males and females and is defined by the failure to achieve a clinical pregnancy after 12 months or more of regular unprotected sexual intercourse [1]. According to the latest survey on infertility, about 48 million couples are affected by infertility [2]. Asthenozoospermia (AZ), which is defined by reduced or absent sperm motility, is one of the main causes of male infertility [3]. In recent years, research has focused on possible factors leading to male infertility and revealed the existence of several cellular and molecular defects that affect sperm production, development and maturation [4,5]. The characterization and identification of the molecular basis of male infertility is challenging due to the high complexity of spermatogenesis, with more than 4000 genes thought to be involved in sperm development [6,7]. Several genes have already been identified and studied as regulators of different aspects of sperm development, maturation and function [8].

AZ has been widely studied, with recent research being focused on the possible mechanisms that could lead to male infertility. Several gene mutations have been associated with AZ [3,9]. According to World Health Organization (WHO) criteria [1], sperm motility can be categorized as: (a) fast progressive motility/grade a (sperm moving ≥ 25 μm/s, either linearly or in a large circle), (b) slow progressive motility/grade b (sperm moving between 5–25 μm/s, either linearly or in a large circle), (c) non-progressive motility/grade c (sperm moving < 5 μm/s, with *in situ* sperm movement with the absence of progression), and (d) immotility/grade d (no active sperm tail movements). AZ can be defined as less than 30% of sperm with grade a + b motility (TPM: total progressive motility), or it can be defined as rapid progressive motility (RPM), with a cut-off of 25% for defining AZ (<25% RPM) [1].

To move properly, sperm need to have the correct sperm flagellum structure, with sperm motility being directly affected by flagella defects, consequently leading to infertility [9]. The microtubule structure at the core of the sperm flagellum that forms its propulsive motor is called the axoneme. The axoneme is a highly complex cylinder structure composed of nine peripheral microtubule doublets, interlinked by the dynein regulatory complex (DRC), and connected by radial spokes (RS) to a single pair of central microtubules [10]. The peripheral doublets include two dynein arms: outer (ODA) and inner (IDA), which are critical for ciliary/flagellar movement [10]. Any structural defect in the axoneme may compromise sperm motility. Besides, morphological anomalies in the sperm structure [11,12], other health complications such as varicocele and infections, lifestyle choices and environmental factors may also cause AZ [9].

The presence of RNA was first described in human sperm nuclei by Pessot et al. [13]. Subsequent investigations, using RT-PCR technology and *in situ* hybridization analysis for sets of RNAs, reported the discovery of RNA in both the sperm head and midpiece [14,15,16]. The amount of RNA in sperm is thought to be quite low, with a single human sperm estimated to contain only 10 to 20 fg [17]. Since the discovery of sperm RNA, an increasing number of studies have demonstrated the accumulation of several RNA types, including both coding and non-coding RNAs, in human sperm with critical functions related to fertility [17,18].

Several transcripts are proposed to be associated with sperm motility. For instance, a study from Bansal and coworkers found that genes *RPL24*, *HNRNPM*, *RPL4*, *PRPF8*, *HTN3*, *RPL11*, *RPL28*, *RPS16*, *SLC25A3*, *C2orf24*, *RHOA*, *GDI2*, *NONO* and *PARK7* were specifically upregulated, whereas genes *HNRNPC*, *SMARCAD1*, *RPS24*, *RPS24*, *RPS27A* and *KIFAP3* were downregulated in AZ patients [19].

Small non-coding RNAs have emerged as key regulators of gene expression in many different cellular pathways, contributing to gene regulation, chromatin structure and inhibition of transposition. Two of the most studied classes are small interfering RNA (siRNAs) and miRNA (miRNA). The presence of these small RNAs has already been demonstrated in spermatogenic cells from both mouse and humans, and revealed a crucial role in spermatogenesis [20,21]. MiRNAs directly interact with partially complementary target sites located in the 3′ untranslated region of target mRNAs, repressing their expression, and thus their main function is gene regulation by the inhibition of translation or by targeting mRNAs for degradation [22]. A well-known example of miRNA is the *miR-34* family, which is a multifunctional miRNA family that has been demonstrated to be important in organ development, organ senescence, stress response, aging, neurodevelopment, signal transduction, and most importantly, human fertility [23,24]. Other examples of miRNAs modulating human fertility include the *MiR-888-3p,* which was found to be significantly overexpressed in AZ patients [25], and *MiR-27a* and *-27b. MiR-27a* and *-27b* negatively regulate the expression of the cysteine-rich secretory protein 2 (*CRISP2*), and downregulation of the CRISP2 protein was significantly associated with low sperm morphology, low progressive motility, and infertility in AZ patients [26]. Thus, the current knowledge about the sperm transcriptome in men and animals has allowed the identification of common differentially expressed genes, which might serve as a molecular diagnostic platform to assist in screening for male infertility causes.

With this work, we aimed to contribute to the knowledge regarding the relationship between the dysregulation of sperm RNA and reduced sperm immotility, by evaluating the expression profiles of a new set of RNA (including mRNA and miRNAs) in human sperm from patients with AZ. In this study, we included genes for which there was evidence of a role in male infertility, based on studies in animal models or from evidence of an action in motile ciliated cells, but with reduced (or absent) information about their role in human sperm cells. The results of this study provide additional findings that contribute to a better understanding of the role of the RNA in sperm function, proving its importance to sperm motility.

## 2. Materials and Methods

### 2.1. Literature Review and Database Search

To select the genes to be included in the study, a literature review was performed in PubMed, using the following key words: asthenozoospermia or sperm immotility, flagellar/ciliary anomalies, genetics of male infertility, sperm RNA and male infertility. Firstly, a list of seventy genes was defined. Thereafter, through inclusion and exclusion parameters (exclusion criteria: existence of multiple studies reporting gene expression analysis in human sperm and high expression in blood cells according to the Human Protein Atlas Database (https://www.proteinatlas.org/, accessed on 1 November 2021); inclusion criteria: a maximum of three reported studies with gene expression analysis in human sperm, high expression in germ cells and ciliated cells), the list was reduced to twenty genes. These include genes *BSCL2*, *CATSPER2*, *CATSPER3*, *CCDC40*, *CFAP43*, *CFAP44*, *CRHR1*, *DRC1*, *HIP1*, *IQCG*, *KRT34*, *LRRC6*, *PLAG1*, *QRICH2*, *RSPH6A*, *SPATA33*, *TEKT2*, *TTC21A*, *USP11* and *ZMYND10*.

### 2.2. Biological Sample Collection

According to the National Law on Medically Assisted Procreation (Law 32/2006) and the National Council for Medically Assisted Procreation guidelines (2018), surplus gametes for research were used under strict individual anonymity and after patients’ written informed consent. The study was approved by the University Hospital Ethics Committee, with authorization number Project: 2019/CE/P017 (266/CETI/ICBAS). Ejaculate samples from 75 patients with reduced RPM (cases) and with normal semen parameters (controls) were obtained at the Centre of Reproductive Genetics Professor Alberto Barros. In all cases, only surplus ejaculates from men undergoing routine spermiogram evaluation were used.

Control nasal cells and peripheral blood were obtained from healthy university volunteers. These cells were derived from our own research RNA bank. Nasal cells were collected by nasal brushing [27] at the hospital by specialized medical personnel. White blood cells were collected from peripheral blood at the hospital by specialized nurse personnel using EDTA-containing tubes (VACUETTE, Porto, Portugal).

Control testicular tissue was derived from our own research RNA bank. This testicular tissue was obtained from excedentary testicular tissue of men with obstructive azoospermia under infertility treatment. Cases with obstructive azoospermia can be used as controls as they present conserved spermatogenesis. Obstructive azoospermic men had normal karyotypes and absence of Y microdeletions and CFTR mutations.

### 2.3. Spermatozoa Isolation

To obtain a population of spermatozoa from the surplus ejaculates, we first separated the seminal plasma from spermatozoa. For that, ejaculate samples were washed with 1 mL of HEPES buffer (Fisher Bio Reagents, Maharashtra, India) and centrifuged at 600× *g* for 10 min at 24 °C, with this step being repeated three times. Supernatants, containing seminal plasma, were stored at −80 °C until use. To eliminate somatic cells from the pellets, these were suspended in somatic cell lysis buffer (SCLB) and incubated on ice for 30 min. SCLB contains 0.1% SDS (TCI Chemicals, Chuo-ku, Tokyo), 0.5% Triton X (Sigma-Aldrich, Saint Louis, MO, USA) and H20-DEPC (Sigma-Aldrich, Saint Louis, MO, USA). Obtainment of a purified spermatozoa pellet was checked by light microscopy to verify elimination of somatic cells. In cases of somatic cell persistence, samples were retreated with SCLB. The pellet was then resuspended in HEPES and centrifuged at 600× *g* for 10 min at 24 °C. The purified spermatozoa were stored at −80 °C until use.

### 2.4. RNA Extraction and cDNA Conversion

The extraction of total RNA from purified spermatozoa was performed using the Single Cell RNA Purification Kit (Cat#. 51800, Norgen, Thorold, ON, Canada). RNA was quantified using the NanoDrop ND-1000 Spectrophotometer (Version 3.3; Life Technologies, Carlsbad, CA, USA). The extracted sperm RNA includes both mRNA and miRNA, as this extraction kit was designed to extract the total RNA from the cell. Part of the extracted RNA was used for cDNA conversion using the High-Capacity cDNA Reverse Transcription Kit (Cat #. 4368814, Applied Biosystems, Waltham, MA, USA), according to the manufacturer’s instructions. The other part of spermatozoa RNA was used for cDNA conversion using the kit NZY First-Strand cDNA Synthesis (MB12501, NZYThech, Lisbon, Portugal) to specifically study miRNA, as it includes an Oligo dT primer, used to produce cDNA from RNA containing a poly(A) tail.

The total RNA from nasal cells, white blood cells and testis, used as positive controls, were extracted with the NZY Total RNA Isolation Kit (MB13402, NZYTech, Lisbon, Portugal), according to the manufacturer’s instructions, and quantified with a NanoDrop spectrophotometer ND-1000 (Version 3.3; Life Technologies, Carlsbad, CA, USA). The synthesis of cDNA from extracted RNA was performed with High-Capacity cDNA Reverse Transcription Kits (Cat #. 4368814, Applied Biosystems, Waltham, Massachusetts, USA), according to the manufacturer’s instructions.

### 2.5. Genes and Primer Design

The reference sequences for the selected genes were retrieved using the National Center for Biotechnology Information (NCBI: http://www.ncbi.nlm.nih.gov/, 1 November 2021) or Ensembl (https://www.ensembl.org/, 1 November 2021). Primer design was performed with Primer Blast (https://www.ncbi.nlm.nih.gov/tools/primer-blast/, 1 November 2021). The parameters used for primer design were as follow: primer length: 18–25 bp; GC content: 40–60%; amplification length: 80–200 bp; and melting temperature: 58–62 °C. Each designed primer pair was tested for the presence of dimer formation using the FastPCR software (version 3.7.7; Institute of Biotechnology, University of Helsinki, Helsinki, Finland) and for their specificity towards the regions of interest using the PrimerBlast tool (NCBI, Bethesda, MD, USA). The primers for mRNA used in this study are listed in the Appendix A.

### 2.6. Polymerase Chain Reaction (PCR)

The PCR technique was used to confirm the expression of the previously selected genes in spermatozoa RNA. As a positive control, a sample in which the presence of each gene was previously confirmed (human testis, human nasal ciliated cells (obtained by nasal brushing) or white blood cells (obtained from peripheral blood)) was included. The reaction mixture (20 μL) contained: 4 μL of FIREPol PCR Master Mix (Cat #. 04-12-00125, Solis BioDyne, Tartu, Estonia), 13 μL of DEPC treated water; 1 μL of each primer (Eurofins Genomics, Ebersberg, Germany) at 10 pmol/μL each, and 1 μL of cDNA at 40 pmol/μL. PCR conditions were optimized for each primer pair and was performed in a PCR T100 thermal cycler (BioRad, Hercules, CA, USA). PCR products were analyzed by 1.5% agarose gel electrophoresis; a mix of TAE 1x SeaKem LE Agarose (Lonza, Basel, Switzerland), and 5 μL/100 mL of GreenSafe Premium (MB13201, NZYTech).

### 2.7. Definition of a List of miRNAs

After gene expression in spermatozoa had been confirmed, a list of miRNAs was generated based on their predicted interaction with the genes previously selected according to the mirDIP database (https://ophid.utoronto.ca/mirDIP/, 1 December 2021) and the Condition-Specific miRNA Targets (CSmiRTar) database (http://cosbi4.ee.ncku.edu.tw/CSmiRTar/, 1 December 2021) (Appendix A). The selection was made using the highest average normalized score, with miRNA with an average normalized score higher than 0.9 and whenever possible supported by more than one database. For genes *CRHR1*, *KRT34* and *QRICH2*, at the time of the study, no miRNA with defined average normalized score was found in these databases, and thus miRNAs from these genes were not included. The miRNA sequences were retrieved using the miRbase (https://www.mirbase.org, 1 December 2021), and primers were designed using the Mirprimer software [28]. The primers for miRNA used in this study are listed in the Appendix A.

### 2.8. Real-Time PCR (RT-PCR)

The mRNA of the selected genes and miRNA was analyzed by quantitative real-time PCR (qPCR), with SYBR green, using the previously designed primers. qPCR was performed in a Bio-Rad CFX96 (Bio-Rad, Hercules, CA, USA) and amplifications were prepared with the NZY qPCR Green (MB22402, NZY-Tech), according to the manufacturer instructions. At the end of each run, a melting curve was completed (from 55 °C to 95 °C with 0.5 °C increments, for 30 s per step) to confirm the specificity of the product. Samples were analyzed in technical triplicates and no-template controls and calibration samples were always included in each 96-well plate setup. For mRNA analysis, the *B2M* and *GAPDH* genes were used as housekeeping genes to normalize gene expression levels. For miRNA analysis, the *miR-30a-5p* and *miR-100-5p* were used to normalize gene expression levels. The fold variation in gene expression levels was calculated following a mathematical model using the formula 2^−ΔΔCt^ [29]. The statistical significance was determined using the non-parametric statistical Kruskal–Wallis test, with α < 0.05. Tests were performed in the GraphPad Prism (version 6.01, GraphPad Software, San Diego, CA, USA).

## 3. Results

### 3.1. Confirmation of the Presence of mRNA in the Purified Spermatozoa

To verify the presence of the twenty selected transcripts in purified spermatozoa, we performed conventional PCR. We observed positive signals from 11 genes: *CATSPER3*, *CFAP44, CRHR1*, *HIP1*, *IQCG*, *KRT34*, *LRRC6*, *QRICH2*, *RSPH6A*, *SPATA33* and *TEKT2* (Figure 1), indicating that those transcripts are present in human sperm. Regarding the remaining genes (*BSCL2*, *CATSPER2*, *CCDC40 CFAP43*, *DRC1*, *PLAG1*, *TTC21A*, *USP11* and *ZMYND10*), we did not observe any positive signal in the same purified spermatozoa population, suggesting that the selected transcripts were not present in sperm or are present at reduced levels and thus were not detected. Ciliated cells, blood cells and testis were used as positive controls.

### 3.2. Analysis of mRNA and miRNA Expression Profiles by Quantitative RT-PCR

After validating the expression of the eleven transcripts in purified spermatozoa through PCR, quantitative RT-PCR was performed. Each transcript was tested in all 33 samples, which were divided into three groups according to the % of RPM: 4 samples of purified spermatozoa with RPM < 15%, 10 samples of purified spermatozoa with RPM between 15–25%, and 19 samples of purified spermatozoa with RPM > 25%. Samples from the RPM < 15% and RPM 15–25% groups were considered our target samples (AZ group) and samples from the RPM > 25% group were considered our control samples. *GAPDH* and *B2M* were used as reference genes. However, as *GAPDH* showed high variability, only *B2M* results was used as a reference (Figure 2).

Using *B2M* as a reference gene, we observed that mRNA from all 11 genes was downregulated in patients with reduced RPM in both subgroups, <15% and 15–25% (Figure 3). In the case of *QRICH2*, *TEKT2*, *KRT34*, *CRHR1*, *HIP1* and *IQCG* it was observed that decreases in the RPM were accompanied by decreasing expression of the transcripts. Genes *QRICH2*, *KRT34*, *TEKT2* and *HIP1* showed the highest fold-decrease, particularly in the RPM < 15% group. In the case of *RSPH6A*, *CATSPER3*, *CFAP44* and *SPATA33*, although the expression of the transcripts was lower in AZ patients, a direct relation between the RPM and the expression level of the target transcript was not observed. For the *LRRC6* transcript, a non-significant reduced expression was observed in the AZ group.

Relatively to miRNAs, two normalizers, miR-30a-5p and miR-100-5p, were used to assess miRNA levels. We studied the expression profiles of miR-3664-5p (*LRRC6*), miR-2110 (*RSPH6A*), miR-4660 (*TEKT2*), miR-492 (*CATSPER3*), miR-4425 (*CFAP44*), miR-4731 (*HIP1*), miR-4514 (*IQCG*) and miR-518c-5p (*SPATA33*). Using both normalizers, all miRNAs were also downregulated in AZ patients when compared to control patients. Except for miR-3664-5p, all miRNAs demonstrated a more accentuated downregulation in patients with the lower RPM (<15%) when compared to the RPM 15–25% group or with the RPM > 25% group. The mir-4660 and mir-2110 were the ones with a high fold-decrease in the group of RPM < 15% (Figure 4).

## 4. Discussion

The goal of the present work was to investigate the expression profiles of a new set of sperm RNA, including mRNA and non-coding RNAs (miRNAs), in human sperm from patients with AZ. After an extensive literature review and expression analyses through PCR, eleven genes associated with sperm immotility were selected, namely, *CATSPER3*, *CFAP44*, *CRHR1*, *HIP1*, *IQCG*, *KRT34*, *LRRC6*, *QRICH2*, *RSPH6A*, *SPATA33* and *TEKT2*. Then, the expression profiles of the mentioned mRNAs were analyzed. Using *B2M* as a reference gene, we observed that mRNA from all 11 genes were downregulated in patients with reduced RPM in both <15% and 15–25% groups (Figure 3).

The expression of the mRNAs *CATSPER3, CFAP44, RSPH6A* and *SPATA33* had the lowest expression in the RPM 15–25% group.

Gene *CATSPER3* is a member of a sperm cation channel-like protein family, named the Cation channel of Sperm (CatSper). *CATSPER3* was proposed to have a role in the acrosome reaction, which is a calcium (Ca^2+^)-dependent secretory event, an essential early step in the fertilization process [30]. High expression levels of *CATSPER3* mRNA have been observed in high-motile human spermatozoa, when compared to low-motile spermatozoa [31]. Further, *CATSPER3* gene expression was observed to be decreased in spermatozoa from patients with AZ and oligoasthenoteratozoospermia (OAT), a condition that includes oligozoospermia (low number of sperm), AZ and teratozoospermia (abnormal sperm shape) [32]. For this gene, our results corroborate these previous findings.

Cilia and flagella associated proteins (CFAPs) play a vital role in the biogenesis of the axoneme and were proposed as responsible for multiple morphological abnormalities of the sperm flagella (MMAF) [33]. Patients with mutations in *CFAP44* (also known as *WDR52*) show a higher rate of spermatozoa with short and absent flagella and a lower motility rate [33,34]. Furthermore, *Cfap44^−^*^/*−*^ male mice also show flagellar immotility, leading to infertility [33]. A recent study analyzed the expression of *CFAP44* and *CFAP44-AS1* (an antisense lncRNA of the CFAP44 gene) in infertile men with sperm motility and morphology defects and observed a significant reduction in the expression of *CFAP44* and *CFAP44-AS1* [35]. Our results also showed a reduced expression in AZ, specifically in the RPM 15–25% group. However, as previous studies analyzing mRNA expression levels from both *CATSPER3* and *CFAP44* only evaluated RPM without analyzing different degrees of severity, we do not know if these studies also noticed any differences regarding the motility rate.

Gene *RSPH6A* (radial spoke head 6 homolog A) is a conserved protein present in eukaryotic cells implicated in flagellar motility [36]. It has been described that *RSPH6A* is testis-specific in humans, being localized in the entire flagellum [36]. In the absence of *RSPH6A*, sperm flagellum elongation may become unstable. The radial spoke is thought to be critical for the axoneme stabilization and for mechano-regulation of flagellar beating [37,38]. Studies have shown that *RSPH6A* interacts with *RSPH1*, *RSPH4A*, *RSPH9* and *RSPH10B*, and is essential for male fertility due to its role in sperm flagellum formation in mice [36]. The present report is thus the first to show that gene *RSPH6A* has a decreased expression in AZ patients.

*SPATA33* (spermatogenesis associated 33) mRNA was shown to be highly expressed in adult mouse testes and was proposed to have a role in kinase signaling during spermatogenesis [39]. *SPATA33* was shown to be important in localizing sperm calcineurin, which is a calcium-dependent phosphatase, to the mitochondria and thus important to sperm motility regulation, with *SPATA33* mutant spermatozoa presenting an inflexible midpiece [40]. This important link might be related to the expression profile of *SPATA33* revealed in the present study.

We could not find studies evaluating gene expression of *SPATA33* and *RSPH6A* in human sperm, and thus the present work is the first to shown *SPATA33* and *RSPH6A* gene expression in human sperm, and a decreased expression in AZ patients. Our results support the role of these genes in sperm motility by showing a reduction in gene expression in AZ patients.

The observation that the lowest mRNA expression was in the RPM 15–25% group in genes *CATSPER3*, *CFAP44*, *RSPH6A* and *SPATA33*, may suggest that these genes may belong to a cascade of genetic events related to sperm motility, but that are not the master players in the process. Thus, a better knowledge of the interactions between these genes and sperm motility is needed to fully understand the present results.

Gene *LRRC6* (leucine-rich repeat (LRR)-containing 6) is mainly expressed in the testis and respiratory epithelial cells and is described as a primary ciliary dyskinesia (PCD)-causing gene. A nonsense mutation that causes the absence of the *LRRC6* protein was reported to result in a defective flagellar structure, specifically in a defective dynein arm, leading to a sperm immotility phenotype [41,42]. The present results for the *LRRC6* transcript showed a non-significant reduced expression in AZ. This suggests that *LRRC6* is potentially specific to a PCD-associated infertility phenotype, and should not be used as a general biomarker of AZ.

The remaining transcripts, *CRHR1*, *HIP1*, *IQCG*, *KRT34*, *QRICH2* and *TEKT2* have shown a reduced expression in patients with AZ, with a more accentuated reduction in patients with an RPM < 15%, which demonstrates a direct correlation between their expression and the RPM in human spermatozoa.

Gene *CRHR1* (corticotropin releasing hormone receptor 1) encodes for a G-protein coupled receptor, which binds to neuropeptides of the corticotropin-releasing hormone (CRH) family, leading to *CRHR1* activation [43]. *CRHR1* has been observed to be downregulated in ciliary cells of a PCD patient, suggesting that *CRHR1* could be involved in the ciliary beating [44]. The *KRT34* gene is a type I cuticular Ha4 keratin, representing a subtype of intermediate filaments (IFs), whose associated proteins are expressed primarily in epithelial cells. It is believed that keratins have functions other than structural support [45]; however, the specific function of *KRT34* remains unclear. Even though there is no direct correlation between *KRT34* and spermatozoa, the expression of *KRT34* in ciliated cells has already been described in a PCD patient and it was suggested that an upregulation of *KRT34* gene may have caused a reduction in its protein expression in ciliated cells [44]. Here, we report for the first time the association of *CRHR1* and *KRT34* with sperm immotility, specifically by showing a reduced expression in patients with AZ, corroborating the hypothesis that *CRHR1* and *KRT34* could be involved in ciliary/flagellar beating. Further studies are needed to understand the exact role of these genes in the motility process.

Gene *HIP1* (huntingtin interacting protein) is an endocytic adaptor protein and a component of clathrin-coated vesicles at the plasma membrane that binds to cytoplasmatic proteins, such as F-actin, tubulin, and huntingtin [46]. Since it binds to both actin and microtubules, *HIP1* might also play a role in cytoskeletal-based vesicular trafficking and/or act as a linking protein between actin and microtubule networks. A study in mice revealed that in the absence of *HIP1*, deformations on the heads, flagella and acrosomes were observed in post-meiotic spermatids, resulting in alterations in sperm motility and count, and fertility [47].

Gene *IQCG* (human IQ motif containing G) is a regulator of calcium signaling that binds to a ubiquitous calcium-binding protein, CaM [48]. Studies in mice have shown that *IQCG* is essential for sperm flagellum formation. *IQCG* might play its role in manchette (a perinuclear mantle of microtubules emanating from the perinuclear ring and linked to the nuclear membrane), which is important for correct sperm nucleus and acrosome elongation, through calcium signaling [49,50]. Besides, IQCG protein interacts with calmodulin in a calcium-dependent manner in the mouse testis, which suggests that IQCG may also participate in sperm calcium signaling [50].

The exact function of gene *QRICH2* (Glutamine rich 2) has not been determined so far, although it is known that it is localized in the sperm flagellum, specifically, QRICH2 is co-localized with microtubule protein α-Tubulin (an axonemal component) [51], as well as with the AKAP4 protein [52]. A recent study proved that *QRICH2* is a functional molecule essential for sperm flagellum development by regulating the genes associated with the flagellum accessory structures, and mutations in this gene resulted in MMAF and male infertility both in men and mice [51]. Another study revealed other homozygous mutations in *QRICH2* in MMAF individuals, confirming that these mutations are rare but are also recurrent in the MMAF phenotype [53]. Furthermore, in bovine males, *QRICH2* mutations cause low sperm concentration and immotile sperm [54]. So, although its exact role is not fully understood, its correlation with MMAF and its importance for the correct formation of the accessory structures of the sperm flagellum has already been proven.

Human *TETK2*, also known as *Tektin-t*, is a membrane protein belonging to the Tektins family of proteins that plays a crucial role in the formation and development of cilia and sperm flagellum [55,56]. Studies have demonstrated that *TEKT2* is localized in the principal piece of spermatozoa and that deficient mice show defects in sperm motility and structural disruption of the IDA since the absence of *TEKT2* in tektin pro filaments may distort the A tubule structure of the microtubule, which can affect the construction of the IDA. This also suggests that *TEKT2* might play a role in the assembly of IDA in the microtubules. On the other hand, the absence of *TEKT2* also affected sperm flagellum formation, leading to ineffective sperm movement, and consequently, to immotile sperm [57].

To the best of our knowledge, we are the first to report the reduced gene expression of *HIP1, IQCG, QRICH2* and *TEKT2* in patients with low sperm motility. Our results further corroborate the previous hypothesis relating these genes to sperm motility, suggesting that they could be used as potential biomarkers of AZ.

Relative to the housekeeping genes, studies that have compared their expression showed that, individually, the expression levels of these genes differ dramatically between tissues. Therefore, the use of more than one control gene for each experiment has been suggested [58,59]. In this study, we used *GAPDH* and *B2M* as reference genes. *GAPDH* (Glyceraldehyde-3-phosphate dehydrogenase) is one of the most common housekeeping genes and its expression is related to the regulation of proliferation, activation status and differentiation [60,61]. Moreover, *GAPDH* was described as one of the most stable reference gene in boar spermatozoa samples [62]. However, our results demonstrated that *B2M* appears more stable when used as a reference gene in human spermatozoa samples than *GAPDH*, which showed a high variability compared to *B2M.* This suggests that *GAPDH* might not be an appropriated reference gene for human spermatozoa gene expression studies and *B2M* should be used instead.

MicroRNA expression profiles play a vital role in regulating gene expression processes. As stated above, this regulation results in the alteration of their target mRNAs translation; additionally, miRNAs are involved in many biological functions, namely, cellular proliferation, differentiation and apoptosis [63]. A single miRNA can have hundreds of target mRNAs, and a mRNA target can bind to one or different miRNAs [22]. This way, miRNAs constitute a powerful regulatory network that controls several targets and physiological processes and diseases [64]. Some miRNAs have specific expression patterns inside the testis, either in different somatic cell types or in germ cells [21]. The presence of miRNAs in mature spermatozoa and seminal plasma and the alterations in their expression profiles in patients with spermatogenic problems demonstrate their potential as biomarkers for diagnosis and classification of male fertility [65,66,67]. These molecules represent a tiny function within the total RNA from samples, which can vary significantly across samples, so its normalization in RT-PCR analyses represents a massive challenge. The Mean-Centering Restricted (MCR) strategy stands out among the developed strategies to measure the expression levels of miRNAs across samples and to identify a normalizer with a uniform and ubiquitous expression [68]. The *miR-100-5p* and *miR-30a-5p* have demonstrated positive results for uniformity and ubiquitous patterns across samples and display a high proximity rate to MCR, which is why they were used as normalizers in the present work [69]. We observed reduced variability among these normalizers, thus corroborating their value in miRNA expression studies of human sperm samples.

It was expected that a downregulation of the mRNAs meant an upregulation of the correspondent miRNAs. So, if all the studied mRNAs are downregulated, we were expecting to observe an upregulation of all miRNAs, meaning that the mRNAs were repressed by the expression of miRNAs. However, the present results showed that all miRNAs were also downregulated in patients with AZ, with a more accentuated reduction in patients from the <15% group, with the exception of *miR-3664-5p. MiR-3664-5p* corresponds to the *LRRC6* transcript, whose results were not statistically significant.

Despite the established role of miRNA in translation repression, studies have emerged reporting that miRNAs can interact with the promoter regions and activate gene expression, a mechanism known as RNA activation [70]. It was reported that *miR-10a* binds in the 5′-UTR region and facilitates translational enhancement rather than repression, with the overexpression of *miR-10a* enhancing ribosomal protein (RP)*,* specifically the RP *Rps16*, *Rps6* and *Rpl9*, synthesis and ribosome biogenesis, and the blockage of miR-10a resulting in the reduced production of these RP proteins [71]. In addition, miR-24-1 was found to be an unconventional mediator for transcriptional gene activation through chromatin remodeling at enhancer regions [72].

So, to justify the absence of an inverted correlation between the expression of the mRNA and the correspondent miRNA, two hypotheses could be proposed. First, it is possible that the miRNAs studied here could, in spermatozoa, act under the mechanism of RNA activation and thus stimulate RNA translation, and that the lower expression observed here would mean a lower expression of their correspondent mRNAs, as was observed. The other hypothesis is related to the powerful regulatory network of miRNA and the fact that some mRNA could have more than one miRNA regulator and vice-versa. The selection of the miRNAs studied here was made through a database search based on highest score of the target miRNA for each mRNA, and thus it is possible that the chosen miRNA might not be the specific target for the spermatozoa mRNAs. Due to alternative splicing, different mRNAs can be found in distinct tissues, and thus it is possible that the same could occur with miRNAs. To validate this hypothesis, further studies should be conducted to analyze other miRNAs that are listed in the databases. Besides, it is also possible that the selected mRNA had multiple target miRNA to regulate their expression, thus the effects of this upregulation were neutralized by the action of the other target miRNA.

## 5. Conclusions

To conclude, our work has provided further insights regarding the role of RNA in sperm function, providing evidence for the first-time that associates genes *CRHR1* and *KRT34* with sperm motility, and that a reduced expression of *HIP1*, *IQCG*, *QRICH2* and *TEKT2* is seen in patients with low sperm motility, with a more accentuated reduction in patients with progressive motility lower than 15%. These results demonstrate a direct correlation between the expression of these genes and the progressive motility rate in human spermatozoa.

We also found that the miRNAs miR-3664-5p (LRRC6), miR-2110 (RSPH6A), miR-4660 (TEKT2), miR-492 (CATSPER3), miR-4425 (CFAP44), miR-4731 (HIP1), miR-4514 (IQCG) and miR-518c-5p (SPATA33) were downregulated in AZ patients, which suggests that this miRNA could act under the mechanism of RNA activation.

Further studies are required to validate this data and determine how these genes and their corresponding miRNAs directly regulate sperm motility, particularly, regarding genes CRHR1 and KRT34 for which information in spermatozoa is scarce.

## Figures and Tables

**Figure 1 genes-13-01291-f001:**
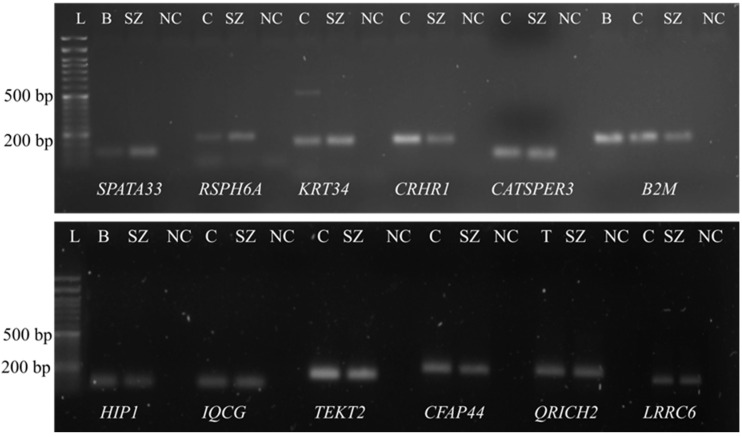
The presence of *SPATA33*, *RSPH6A*, *KRT34*, *CRHR1*, *CATSPER3* (**in the upper gel**), *HIP1*, *IQCG*, *LRCC6*, *QRICH2*, *TEKT2* and *CFAP44* (**in the lower gel**) transcripts in human ejaculated spermatozoa identified by PCR and running on a 1.5% agarose gel. *B2M* was used as control for the different tissues. L, DNA Ladder; B, blood; C, ciliated cells; T, testis; SZ, purified spermatozoa; NC, negative control.

**Figure 2 genes-13-01291-f002:**
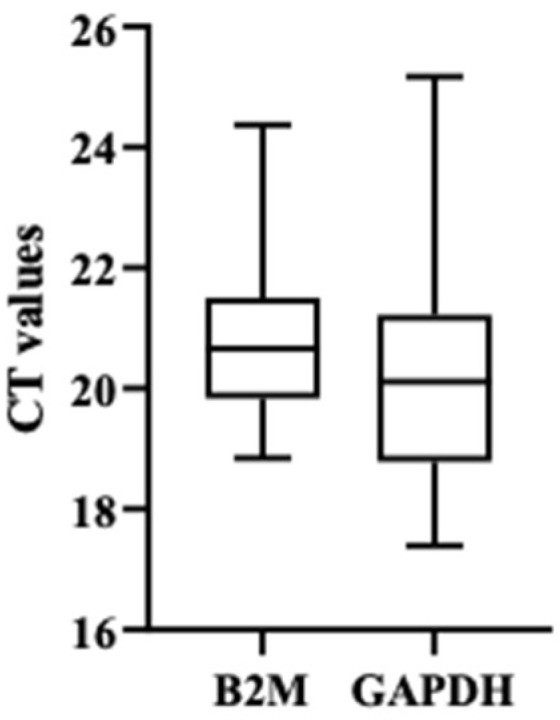
Box plot of CT values of *B2M* and *GAPDH* genes.

**Figure 3 genes-13-01291-f003:**
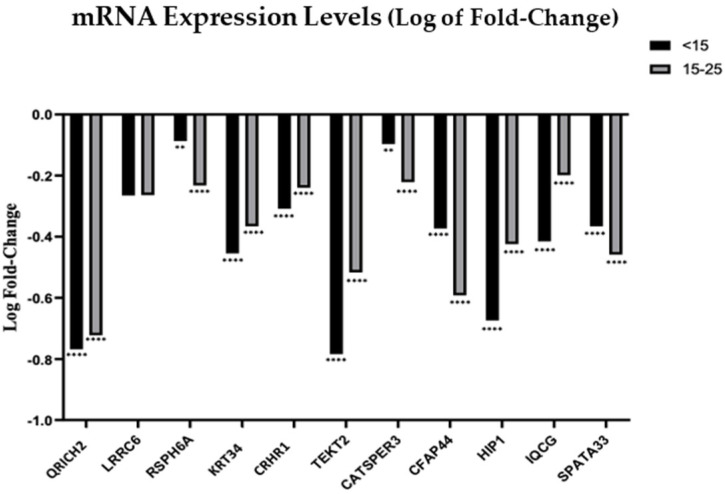
Log of fold change of the mRNA expression levels in spermatozoa from patients with RPM < 15% and RPM between 15–25%. Statistical significance was determined using the Kruskal–Wallis test, with α < 0.05. ** *p* < 0.01, and **** *p* < 0.0001. *B2M* was used as reference gene.

**Figure 4 genes-13-01291-f004:**
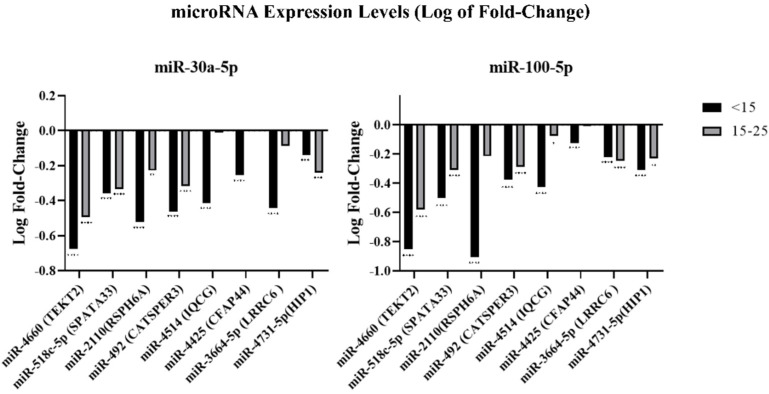
Log of fold change of the miRNA expression levels in spermatozoa from patients with RPM < 15% and RPM between 15–25%. Statistical significance was determined using the Kruskal–Wallis test, with α < 0.05. * *p* < 0.05, ** *p* < 0.01, *** *p* < 0.001 and **** *p* < 0.0001. *miR-30a-5p* (**left**) and *miR-100-5p* (**right**) were used as reference genes.

## Data Availability

The authors confirm that the data supporting the findings of this study are available within the article and its Appendix A.

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
