# Peer review of "Further Insights on RNA Expression and Sperm Motility"

_genes, 2022, doi:10.3390/genes13071291_

Round 1
Reviewer 1 Report
Comments:
Line 93: "Thus, the current knowledge about the sperm transcriptome in men and animals suggests that sperm RNA profiles could be used as a genetic fingerprint of normal fertile males and as a molecular diagnostic platform for male infertility."
Comment: Abnormal sperm RNA profiles could be used to aid in screening for male infertility, but I doubt screening for normal RNA profiles would be an accurate way for assessing normal male infertility by itself.
Line 96: "With this work, we aimed to contribute to increase knowledge regarding the relationship between dysregulation of sperm RNA and reduced sperm immotility, by evaluating the expression profiles of a new set of RNA (including mRNA and miRNAs) in human sperm from patients with AZ. The results of this study provide additional findings towards a better understanding of the role of the RNA in sperm function, proving its importance to sperm motility."
Comment: you should have a hypothesis why you screened the RNAs you chose to screen? please summarize this rationale in the introduction.
Line 137: primer design. Please list who made your primers in methods section
Line 149: "using the Single Cell RNA Purification Kit (Norgen, Thorold, Canada)" please list kit number or Cat#
Line 154: "cDNA Reverse Transcription Kit (Applied Biosystems, California, USA)" please list kit number or Cat#
Line 156: "kit NZY First-Strand cDNA Synthesis (NZYThech, Lisbon, Portugal)" please list kit number or Cat#
Line 162: Methods section. Please outline where and how DNA/RNA was collected from your positive control tissues: blood/ciliated cells (what type of ciliated cell)/and testis.
Line 212: what was used as a negative control. please add this to methods section.
Figures 3 & 4. Why don't you also graph your PM >25% group?
Line 217: "and 19 samples of purified spermatozoa with PM >25% were considered our control samples."
Why use 25% as your cut off point? your introduction states "AZ is defined as less than 30% of sperm with grade a+b motility (PM: progressive motility)". Why didn't you use 30% as your cut-off point? Could your controls have AZ samples?
Minor suggested corrections
Line 14: change "the role of sperm RNA in regulation of male sperm motility" to "the role of sperm RNA in regulating sperm motility"
Line 15: following needs rewriting for clarity "Consequently, here we aim to give contribute to increase knowledge regarding the expression of sperm RNA and its relationship with sperm motility"
Line 18: "and its highest score corresponding target microRNA"
What does "it" refer to? (change to "their"?)
What score are you referring to?
Line 20: change "in patients with reduced sperm motility" to "in asthenozoospermic patients compared to controls"
line 21: change "motility" to "sperm motility"
Line 21: change "has provided" to "provides"
Line 22: change "the role of RNA in sperm function, proving its importance to sperm motility" to "the role of RNA in regulating sperm motility"
Line 23: change "Further studies are important to understand" to "Further studies are required to determine"
Line 23: change "and its corresponding" to "and their corresponding"
Line 24 change "act in sperm motility" to "regulate sperm motility"
Line 24 change "particularly, KRT34 and CRHR1 genes of which information in sperm is scarce." to "particularly KRT34 and CRHR1 which have not previously been seen to play a significant role in regulating sperm motility."
Line 42: change "Several gene mutations were reported associated with AZ"
to "Several gene mutations have been associated with AZ"
Line 43: change "According to WHO criteria," to "According to WHO criteria [1], "
Line 43: change "can be categorized in:" to "can be categorized as: "
Line 44: change "(25 μm/s, sperm moving actively, either linearly or in a large circle)" to "(sperm moving ≥25 μm/s, either linearly or in a large circle)"
Line 45: change "(5 to < 25 μm/s, sperm moving actively, either linearly or in a large circle)" to "(sperm moving between 5 - 25 μm/s, either linearly or in a large circle)"
Line 47: change (< 5 μm/s, sperm tail in situ movements with absence of progression)" to "(sperm moving < 5 μm/s, with in situ sperm movement with absence of progression)"
Line 48: missing ")"
Line 51: change "defects in flagella" to "flagella defects"
Line 52: change "The sperm flagellum contains a microtubular structure named axoneme that is the sperm propulsive motor" to "The microtubule structure at the core of the sperm flagellum forming its propulsive motor is called the axoneme."
Line 56: change "...microtubules." to "...microtubules [10]."
Line 63: change "The presence of RNA was firstly described in the human sperm nucleus [13]." to "The presence of RNA was first described in human sperm nuclei by Pessot et al [13]."
Line 65: change "allowed the discovery of several transcripts to be present in both sperm head and midpiece" to "reported the discovery of RNA in both sperm head and midpiece [14-16]."
Line 66: change "although at low amounts, being estimated that a single human sperm contains about 10 to 20 fg of RNA" to "The amount of RNA in sperm is thought quite low, with a single human sperm estimated to contain only 10 to 20 fg"
line 68: change "populations' to "types"
Line 71: change "Several transcripts were" to "Several RNA transcripts are"
Line 85: change "which is a multifunctional miRNA family that have been demonstrated to be important in organ development, organ senescence, stress response, aging, neurodevelopment and signal transduction [23]. Besides, the miR-34 family have proven to have critical roles in fertility in humans [24]" to
"which is a multifunctional miRNA family that has been demonstrated to be important in organ development, organ senescence, stress response, aging, neurodevelopment, signal transduction, and most importantly human fertility [23, 24]."
Line 88: change "Another examples are the" to "Other examples of MicroRNAs modulating human fertility include"
Line 125: "(CGR)" No need to define acronym you only use once.
Line 132: change "Obtainment of a purified spermatozoa pellet was check by" to "Spermatozoa pellet purity was checked by"
Line 151: change "includes" to "included"
Line 153: change "was used to cDNA conversion" to "was used to generate cDNA"
Line 155: change "Other part of sperm RNA was used to cDNA conversion using" to "The remaining sperm RNA was used to generate cDNA using"
Line 171: change "After defined the group of genes expressed in spermatozoa" to "After gene expression in spermatozoa had been confirmed"
Line 171: change "a list of microRNAs that are predicted to interact with some of the genes previously selected according the databases mirDIP database" to "a list of microRNAs was generated based on their predicted interaction with the genes previously selected according to the mirDIP database"
Line 178: change "found in databases" to "found in these databases"
Line 179: change "sequence was" to "sequences were"
Line 220: change "However, GAPDH showed a high variability and only B2M results were assumed" to ""However, as GAPDH showed high variability only B2M results was used as a reference"
Line 237: "to assess gene expression" shouldn't that be "to assess miRNA levels"
Line 257: change "that all mRNA" to "that mRNA from all 11 genes"
Lines 259-260: isn't this just repeating the previous sentence?
Line 271-272: "were proposed as responsible for multiple morphological abnormalities of 271 the sperm flagella (MMAF)." needs citation at end of this sentence
Line 284: "eukaryotic cells implicated in flagellar motility." needs citation at end of this sentence
Line 354: "although it is known that is localized in the sperm flagellum. " needs citation at end of this sentence
Line 389: change "In what concerns microRNAs, their expression profile is considered as a key in gene expression regulation processes, being recognized as vital regulatory factors." to "microRNA expression profiles play a vital role in regulating gene expression processes"
Line 393: "differentiation and apoptosis." needs citation at end of this sentence
Line 397: "in different somatic cell types or in germ cells." needs citation at end of this sentence
Line 426: change "correlation the expression" to "correlation between the expression"
Line 444: change "proving for the first time evidence that associate genes CRHR1 and KRT34 to sperm motility and report, also for the first time, a reduced expression of HIP1, IQCG, QRICH2 and TEKT2 in patients with low sperm motility, with a more accentuated reduction in patients with progressive motility lower than 15%." to "providing for the first-time evidence that associates genes CRHR1 and KRT34 with sperm motility, and that a reduced expression of HIP1, IQCG, QRICH2 and TEKT2 is seen in patients with low sperm motility, with a more accentuated reduction in patients with progressive motility lower than 15%."
Line 450: change "We also firstly report that the" to "We also find that the"
Line 454: change "Certainly, these data need to be more deeply investigated. Further studies with a higher number of patients can further validate this data and studies using animal models are important for future research on the functions of these genes and how these genes and its corresponding microRNA act in sperm motility, particularly, regarding genes CRHR1 and KRT34 of which information in spermatozoa is scare." to "Further studies are required to validate this data and determine how these genes and their corresponding microRNAs directly regulate sperm motility, particularly, regarding genes CRHR1 and KRT34 of which information in spermatozoa is scare."
Author Response
Dear reviewer,
We are grateful for your precious comments to improve our manuscript. We have done our best to correct as you have suggested. Here are our responses to your comments. Besides, we included in a word as well to be easier for the reviewer.
Line 93: "Thus, the current knowledge about the sperm transcriptome in men and animals suggests that sperm RNA profiles could be used as a genetic fingerprint of normal fertile males and as a molecular diagnostic platform for male infertility."
Comment: Abnormal sperm RNA profiles could be used to aid in screening for male infertility, but I doubt screening for normal RNA profiles would be an accurate way for assessing normal male infertility by itself.
Answer: Indeed, we were too optimistic. We will reformulate as “Thus, the current knowledge about the sperm transcriptome in men and animals have allowed the identification of common differentially expressed genes, which might serve as molecular diagnostic platform to assist in screening for male infertility causes."
Line 96:"With this work, we aimed to contribute to increase knowledge regarding the relationship between dysregulation of sperm RNA and reduced sperm immotility, by evaluating the expression profiles of a new set of RNA (including mRNA and miRNAs) in human sperm from patients with AZ. The results of this study provide additional findings towards a better understanding of the role of the RNA in sperm function, proving its importance to sperm motility."
Comment: you should have a hypothesis why you screened the RNAs you chose to screen? Please summarize this rationale in the introduction.
Answer: We included in the method section the rationale. Briefly, we made a list with genes that are likely to be related with sperm motility by searching on Pubmed using key words: asthenozoospermia or sperm immotility, flagellar/ciliary anomalies, genetics of male infertility, sperm RNA and male infertility. We narrow our research to genes in which there were evidence a role on male infertility, namely by the existence of studies in animal models or evidence of an action in motile ciliated cells, but reduced (or absent) information in human sperm cells.
As the Reviewer suggested, we have included in introduction the following sentence “We have included in this study genes in which there were evidence a role on male infertility, obtained from studies in animal models or from evidence of an action in motile ciliated cells, but reduced (or absent) information in human sperm cells”
Line 137: primer design. Please list who made your primers in methods section
Answer: The primers were designed by us. Briefly, the reference sequences for the selected genes were retrieved using the National Center for Biotechnology Information (NCBI: http://www.ncbi.nlm.nih.gov/) or Ensembl (https://www.ensembl.org/), and Primer3 plataform (https://bioinfo.ut.ee/primer3/) was used for design primers. Parameters used for primer design were as follow: primer length: 18-25 bp; GC content: 40-60%; amplification length: 80-250 bp; and melting temperature: 58-66 C°. Each designed primer pair was tested for the presence of dimer formation using the FastPCR software (version 3.7.7; Institute of Biotechnology, University of Helsinki, Finland) and for their specificity towards the regions of interest using the PrimerBlast tool (NCBI, Bethesda, USA).
Line 149: "using the Single Cell RNA Purification Kit (Norgen, Thorold, Canada)" please list kit number or Cat#
Answer: As suggested we have included the Cat#. 51800
Line 154: "cDNA Reverse Transcription Kit (Applied Biosystems, California, USA)" please list kit number or Cat#
Answer: As suggested, we have included the Cat #. 4368814
Line 156: "kit NZY First-Strand cDNA Synthesis (NZYThech, Lisbon, Portugal)" please list kit number or Cat#
Answer: As suggested, we have included the kit number MB12501
Line 162: Methods section. Please outline where and how DNA/RNA was collected from your positive control tissues: blood/ciliated cells (what type of ciliated cell)/and testis.
Answer: As suggested, we have included in the Methods section the following information:
2.2. Biological sample collection
According to the National Law on Medically Assisted Procreation (Law 32/2006) and the National Council for Medically Assisted Procreation guidelines (2018), surplus gametes for research were used under strict individual anonymity and after patient written informed consent. The study was approved by the University Hospital Ethics Committee, with authorization number Project: 2019/CE/P017 (266/CETI/ICBAS). Ejaculate samples from 75 patients with reduced rapid progressive motility (cases) and with normal semen parameters (controls) were obtained at the Centre of Reproductive Genetics Prof. Alberto Barros. In all cases, only surplus ejaculates from men undergoing routine spermiogram evaluation were used.
Control nasal cells and peripheral blood were obtained from healthy university volunteers. These cells derived from own research RNA bank. Nasal cells were collected by nasal brushing (Ref de artigo nosso) at the hospital under specialized medical personnel. White blood cells were collected from peripheral blood at the hospital under specialized nurse personnel using EDTA containing tubes (VACUETTE, Porto, Portugal).
Control testicular tissue derived from own research RNA bank. This testicular tissue was obtained from excedentary testicular tissue of men with obstructive azoospermia under infertility treatments. Cases with obstructive azoospermia can be used as controls as they present conserved spermatogenesis. Obstructive azoospermic men had normal karyotypes and absence of Y microdeletions and CFTR mutations.
2.3 Spermatozoa isolation
To obtain a population of spermatozoa from the surplus ejaculates, we first separate the seminal plasma from spermatozoa. For that, ejaculate samples were washed with 1 ml of HEPES buffer (Fisher Bio Reagents, Maharashtra, India) and centrifuged at 600 × g for 10 min at 24 C°, with this step being repeated three times. Supernatants, containing seminal plasma, were stored at -80ºC until use. To eliminate somatic cells from the pellets, these were suspended in somatic cell lysis buffer (SCLB) and incubated on ice for 30 min. SCLB contains 0.1% SDS (TCI Chemicals, California, USA), 0.5% Triton X (Sigma-Aldrich, Missouri, USA) and H20-DEPC (Sigma-Aldrich). Obtainment of a purified spermatozoa pellet was checked by light microscopy to verify elimination of somatic cells. In cases of somatic cell persistence, samples were retreated with SCLB. The pellet was then resuspended in HEPES and centrifuged at 600 × g for 10 min at 24 C°. The purified spermatozoa were stored at -80ºC until use.
2.4. RNA extraction and cDNA conversion
The extraction of total RNA from purified spermatozoa was performed using the Single Cell RNA Purification Kit (Cat#. 51800, Norgen, Thorold, Canada). RNA was quantified using the NanoDrop ND-1000 Spectrophotometer (Version 3.3; Life Technologies). The extracted sperm RNA includes both mRNA and miRNA, as this extraction kit is designed to extract the total RNA from the cell. Part of the extracted RNA was used to cDNA conversion using the High-Capacity cDNA Reverse Transcription Kit (Cat #. 4368814, Applied Biosystems, California, USA), according to the manufacturer instructions. Other part of spermatozoa RNA was used to cDNA conversion using the kit NZY First-Strand cDNA Synthesis (MB12501, NZYThech, Lisbon, Portugal) to specifically study microRNA, as it includes an Oligo dT primer, used to produce cDNA from RNA containing a poly(A) tail.
The total RNA from nasal cells, white blood cells and testis, used as positive controls, were extracted with the NZY Total RNA Isolation Kit (MB13402, NZYTech, Lisbon, Portugal), according to manufacturer instructions, and quantified with a NanoDrop spectrophotometer ND-1000. The synthesis of cDNA, from extracted RNA, was performed with High-Capacity cDNA Reverse Transcription Kits (Applied Biosystems), according to the manufacturer instructions.
Line 212: what was used as a negative control? Please add this to the Methods section.
Answer: As negative control, we have used a blank sample with no RNA to infer contamination or primer dimer.
Figures 3 & 4. Why don't you also graph your PM >25% group?
Answer: In figures 3 and 4 we show the log of fold change of the mRNA and miRNA expression levels, respectively. This fold change was applied to obtain the up-regulated vs down-regulated mRNA/miRNA in relation to control group (that is the PM >25% group), therefore this group is not represented. To calculate the fold change we divided the average 2^(delta ct) values from the low PM groups by the average 2^(delta ct) values from the > 25 PM group. Then, we did the log of the previous value.
Line 217: "and 19 samples of purified spermatozoa with PM >25% were considered our control samples." Why use 25% as your cut off point? Your Introduction states "AZ is defined as less than 30% of sperm with grade a+b motility (PM: progressive motility)". Why didn't you use 30% as your cut-off point? Could your controls have AZ samples?
Answer: We are sorry for this error. In the WHO edition of semen analysis, besides total Progressive Motility (PM), with a cut off of 30% for defining asthenozoospermia (< 30% PM), it can also be used rapid progressive motility (RPM), with a cut off of 25% for defining asthenozoospermia (< 25% RPM). Total PM includes slow progressive motility (SPM) and RPM. However, it is believed that only spermatozoa with RPM have the capacity to move along the female genital tract (vaginal channel, cervix ostium, uterine cavity, tube cavity) and penetrate oocyte vestments (cumulus follicular cells, zona pellucida), thus enabling full fertilization. Although all of our normozoospermic samples had a total PM > 30%, to avoid misunderstandings we included the explanation regarding the TPM and RPM and corrected the text PM to RPM.
Minor suggested corrections
Line 14: change "the role of sperm RNA in regulation of male sperm motility" to "the role of sperm RNA in regulating sperm motility"
Answer: We reformulated as you suggested.
Line 15: following needs rewriting for clarity "Consequently, here we aim to give contribute to increase knowledge regarding the expression of sperm RNA and its relationship with sperm motility"
Answer: Reformulate as follows “Consequently, here we aim to give contribute to increase knowledge regarding the expression of sperm RNA and ultimately to give further insights on its relationship with sperm motility.”
Line 18: "and its highest score corresponding target microRNA"
What does "it" refer to? (change to "their"?)
Answer: Reformulated as follows “We investigated by real-time PCR the expression of a group of mRNAs (CATSPER3, CFAP44, CRHR1, HIP1, IQCG KRT34, LRRC6, QRICH2, RSPH6A, SPATA33 and TEKT2) and the highest score corresponding target microRNA for each mRNA, in asthenozoospermic and normozoospermic individuals.
What score are you referring to?
Answer: We are referring to the average normalized score obtained in the databases mirDIP database (https://ophid.utoronto.ca/mirDIP/) and CSmiRTar database – Condition-Specific miRNA Targets database - (http://cosbi4.ee.ncku.edu.tw/CSmiRTar/). Due to the existence of several microRNAs for each mRNA, to select the microRNA for this study, we have selected the one with a higher average normalized score, as is referred in methods section.
Line 20: change "in patients with reduced sperm motility" to "in asthenozoospermic patients compared to controls"
Answer: Thank you. We reformulated as you suggested.
Line 21: change "motility" to "sperm motility"
Answer: Thank you. We reformulated as you suggested.
Line 21: change "has provided" to "provides"
Answer: Thank you. We reformulated as you suggested.
Line 22: change "the role of RNA in sperm function, proving its importance to sperm motility" to "the role of RNA in regulating sperm motility"
Answer: Thank you. We reformulated as you suggested.
Line 23: change "Further studies are important to understand" to "Further studies are required to determine"
Answer: Thank you. We reformulated as you suggested.
Line 23: change "and its corresponding" to "and their corresponding"
Answer: Thank you. We reformulated as you suggested.
Line 24: change "act in sperm motility" to "regulate sperm motility"
Answer: Thank you. We reformulated as you suggested.
Line 24: change "particularly, KRT34 and CRHR1 genes of which information in sperm is scarce." to "particularly KRT34 and CRHR1, which have not previously been seen to play a significant role in regulating sperm motility."
Answer: Thank you. We reformulated as you suggested.
Line 42: change "Several gene mutations were reported associated with AZ"
to "Several gene mutations have been associated with AZ"
Answer: Thank you. We reformulated as you suggested.
Line 43: change "According to WHO criteria," to "According to WHO criteria [1], "
Answer: Thank you. We reformulated as you suggested.
Line 43: change "can be categorized in:" to "can be categorized as: "
Answer: Thank you. We reformulated as you suggested.
Line 44: change "(25 μm/s, sperm moving actively, either linearly or in a large circle)" to "(sperm moving ≥25 μm/s, either linearly or in a large circle)"
Answer: Thank you. We will reformulate as you suggest.
Line 45: change "(5 to < 25 μm/s, sperm moving actively, either linearly or in a large circle)" to "(sperm moving between 5 - 25 μm/s, either linearly or in a large circle)"
Answer: Thank you. We reformulated as you suggested.
Line 47: change (< 5 μm/s, sperm tail in situ movements with absence of progression)" to "(sperm moving < 5 μm/s, with in situ sperm movement with absence of progression)"
Answer: Thank you. We reformulated as you suggested.
Line 48: missing ")"
Answer: Thank you.
Line 51: change "defects in flagella" to "flagella defects"
Answer: Thank you. We reformulated as you suggested.
Line 52: change "The sperm flagellum contains a microtubular structure named axoneme that is the sperm propulsive motor" to "The microtubule structure at the core of the sperm flagellum forming its propulsive motor is called the axoneme."
Answer: Thank you. We reformulated as you suggested.
Line 56: change "...microtubules." to "...microtubules [10]."
Answer: Thank you. We reformulated as you suggested.
Line 63: change "The presence of RNA was firstly described in the human sperm nucleus [13]." to "The presence of RNA was first described in human sperm nuclei by Pessot et al [13]."
Answer: Thank you. We reformulated as you suggested.
Line 65: change "allowed the discovery of several transcripts to be present in both sperm head and midpiece" to "reported the discovery of RNA in both sperm head and midpiece [14-16]."
Answer: Thank you. We reformulated as you suggested.
Line 66: change "although at low amounts, being estimated that a single human sperm contains about 10 to 20 fg of RNA" to "The amount of RNA in sperm is thought quite low, with a single human sperm estimated to contain only 10 to 20 fg"
Answer: Thank you. We reformulated as you suggested.
Line 68: change "populations' to "types"
Answer: Thank you. We reformulated as you suggested.
Line 71: change "Several transcripts were" to "Several RNA transcripts are"
Answer: Thank you. We reformulated as you suggested.
Line 85: change "which is a multifunctional miRNA family that have been demonstrated to be important in organ development, organ senescence, stress response, aging, neurodevelopment and signal transduction [23]. Besides, the miR-34 family have proven to have critical roles in fertility in humans [24]" to "which is a multifunctional miRNA family that has been demonstrated to be important in organ development, organ senescence, stress response, aging, neurodevelopment, signal transduction, and most importantly human fertility [23, 24]."
Answer: Thank you. We reformulated as you suggested.
Line 88: change "Another examples are the" to "Other examples of MicroRNAs modulating human fertility include"
Answer: Thank you. We reformulated as you suggested.
Line 125: "(CGR)" No need to define acronym you only use once
Answer: Thank you. We will remove the acronym.
Line 132: change "Obtainment of a purified spermatozoa pellet was check by" to "Spermatozoa pellet purity was checked by"
Answer: Thank you. We reformulated as you suggested.
Line 151: change "includes" to "included"
Answer: Thank you. We reformulated as you suggested.
Line 153: change "was used to cDNA conversion" to "was used to generate cDNA"
Answer: Thank you. We reformulated as you suggested.
Line 155: change "Other part of sperm RNA was used to cDNA conversion using" to "The remaining sperm RNA was used to generate cDNA using"
Answer: Thank you. We reformulated as you suggested.
Line 171: change "After defined the group of genes expressed in spermatozoa" to "After gene expression in spermatozoa had been confirmed"
Answer: Thank you. We reformulated as you suggested.
Line 171: change "a list of microRNAs that are predicted to interact with some of the genes previously selected according the databases mirDIP database" to "a list of microRNAs was generated based on their predicted interaction with the genes previously selected according to the mirDIP database"
Answer: Thank you. We reformulated as you suggested.
Line 178: change "found in databases" to "found in these databases"
Answer: Thank you. We reformulated as you suggested.
Line 179: change "sequence was" to "sequences were"
Answer: Thank you. We reformulated as you suggested.
Line 220: change "However, GAPDH showed a high variability and only B2M results were assumed" to ""However, as GAPDH showed high variability only B2M results was used as a reference"
Answer: Thank you. We reformulated as you suggested.
Line 237: "to assess gene expression" shouldn't that be "to assess miRNA levels"
Answer: Thank you. Indeed, it should. We have corrected that.
Line 257: change "that all mRNA" to "that mRNA from all 11 genes"
Answer: Thank you. We reformulated as you suggested.
Lines 259-260: isn't this just repeating the previous sentence?
Answer: We reformulated as “The expression of the mRNAs CATSPER3, CFAP44, RSPH6A and SPATA33 have the lowest expression in the PM 15-25% group”.
Line 271-272: "were proposed as responsible for multiple morphological abnormalities of 271 the sperm flagella (MMAF)."needs citation at end of this sentence
Answer: A reference was included.
Line 284: "eukaryotic cells implicated in flagellar motility." needs citation at end of this sentence.
Answer: A reference was included.
Line 354: "although it is known that is localized in the sperm flagellum. " needs citation at end of this sentence
Answer: We reformulated as ” The exact function of gene QRICH2 (Glutamine rich 2) has not been determined so far, although it is known that is localized in the sperm flagellum, specifically QRICH2 is co-localized with microtubule protein α-Tubulin (an axonemal component) [50], as well as, with the AKAP4 protein [51].”
Line 389: change "In what concerns microRNAs, their expression profile is considered as a key in gene expression regulation processes, being recognized as vital regulatory factors." to "microRNA expression profiles play a vital role in regulating gene expression processes"
Answer: Thank you. We reformulated as you suggested.
Line 393: "differentiation and apoptosis." needs citation at end of this sentence.
Answer: A reference was included.
Line 397: "in different somatic cell types or in germ cells." needs citation at end of this sentence.
Answer: A reference was included.
Line 426: change "correlation the expression" to "correlation between the expression"
Answer: Thank you. We reformulated as you suggested.
Line 444: change "proving for the first time evidence that associate genes CRHR1 and KRT34 to sperm motility and report, also for the first time, a reduced expression of HIP1, IQCG, QRICH2 and TEKT2 in patients with low sperm motility, with a more accentuated reduction in patients with progressive motility lower than 15%." to "providing for the first-time evidence that associates genes CRHR1 and KRT34 with sperm motility, and that a reduced expression of HIP1, IQCG, QRICH2 andTEKT2 is seen in patients with low sperm motility, with a more accentuated reduction in patients with progressive motility lower than 15%."
Answer: Thank you. We reformulated as you suggested.
Line 450: change "We also firstly report that the" to "We also find that the"
Answer: Thank you. We reformulated as you suggested.
Line 454: change "Certainly, these data need to be more deeply investigated. Further studies with a higher number of patients can further validate this data and studies using animal models are important for future research on the functions of these genes and how these genes and its corresponding microRNA act in sperm motility, particularly, regarding genes CRHR1 and KRT34 of which information in spermatozoa is scare." to "Further studies are required to validate this data and determine how these genes and their corresponding microRNAs directly regulate sperm motility, particularly, regarding genes CRHR1 and KRT34 of which information in spermatozoa is scare."
Answer: Thank you. We reformulated as you suggested.

Reviewer 2 Report
The authors of the study entitled "Further insights on RNA expression and sperm motility " are trying to explore the role of RNA in sperm function. Interestingly, this study also first reported a few gene candidates in infertile human patients. This would be very helpful to draw a differential gene expression sketch of infertility in humans clinically. The study is well designed and well-written. This manuscript is suitable for publication with revision.
Comments:
1. Lines 52-58, looks unnecessarily incorporated (generic lines). Please remove these lines or provide them briefly.
2. It is not clear which ciliated cells have been used as a positive control.
3. Figure 3.1. Lower panel, It seems that onwards 3rd genes it's not the same gel images with ladder. Please provide new images having a ladder in the same gel with all the genes.
4. Figure 3.1: It is not clear in both the images why few genes’ lanes have a comparison between blood-spermatozoa; Testis-spermatozoa and ciliated cells-spermatozoa. (Different positive controls for a few genes)
5. Author would provide clarification about the expression profile of SPATA33, HIP1, and QUICH2 genes in ciliated cells used in this study.
6. This will be helpful if the author provides the size of ladder bands (kb) on the left side of the gel image.
7. Author would re-assign the name of the lane, especially as B- Blood; NC-negative control.
8. Author would change the title of the 3.1 figure.
9. Manuscript is well written; However, minor language improvement is needed throughout the manuscript.
Author Response
Dear Reviewer,
We acknowledge your precious comments, which were very important to improve our manuscript.
We hope to fullfil all your doubts and make the manuscript more clear.
Above are the answers to your comments. We have also include a word to be easier for the reviewer.
Comments:
1-Lines 52-58, looks unnecessarily incorporated (generic lines). Please remove these lines or provide them briefly.
Answer: We reduced as follows “The axoneme is a highly complex cylinder structure composed by nine peripheral microtubule doublets, interlinked by the dynein regulatory complex (DRC), and connected by radial spokes (RS) to a single pair of central microtubules. The peripheral doublets include two dynein arms: outer (ODA) and inner (IDA), which are critical for ciliary/flagellar movement”
2-It is not clear which ciliated cells have been used as a positive control.
Answer: we perform external hospital diagnosis for primary ciliary dyskinesia. In this setting, ciliated cells are analyzed through high-speed video microscopy (HSVM) and transmission electron microscopy. After HSVM analysis, excedentary ciliated cells and their retrieved RNA are frozen for molecular diagnosis and research. During research studies, we also collect ciliated cells from healthy university volunteers. Ciliated cells are obtained through nasal brushing. Another research field in our group is dedicated to reproductive medicine, where excedentary testicular tissue and their retrieved RNA is frozen for diagnosis and research studies.
This is now more clarified in the Methods section:
Control nasal cells and peripheral blood were obtained from healthy university volunteers. These cells derived from own research RNA bank. Nasal cells were collected by nasal brushing (27) at the hospital under specialized medical personnel. White blood cells were collected from peripheral blood at the hospital under specialized nurse personnel using EDTA containing tubes (VACUETTE, Porto, Portugal).
Control testicular tissue derived from own research RNA bank. This testicular tissue was obtained from excedentary testicular tissue of men with obstructive azoospermia under infertility treatments. Cases with obstructive azoospermia can be used as controls as they present conserved spermatogenesis. Obstructive azoospermic men had normal karyotypes and absence of Y microdeletions and CFTR mutations.
3-Figure 3.1. Lower panel. It seems that onwards 3rd genes it's not the same gel images with ladder. Please provide new images having a ladder in the same gel with all the genes.
Answer: We corrected the image as you suggest.
4-Figure 3.1: It is not clear in both the images why few genes’ lanes have a comparison between blood-spermatozoa; Testis-spermatozoa and ciliated cells-spermatozoa. (Different positive controls for a few genes)
Answer: We used as positive control the sample that in the human protein atlas database is referred to have a higher expression, which is why we included blood, ciliated or testis samples.
5-Author would provide clarification about the expression profile of SPATA33, HIP1, and QUICH2 genes in ciliated cells used in this study.
Answer: For these genes we have not used ciliated cells as positive control, but white blood cells or testis.
6-This will be helpful if the author provides the size of ladder bands (kb) on the left side of the gel image.
Answer: We corrected the image as you suggest.
7-Author would re-assign the name of the lane, especially as B- Blood; NC-negative control.
Answer: We reformulated as you suggested.
8-Author would change the title of the 3.1 figure.
Answer: Sorry but we do not have figure 3.1, but we assume that revisor are referring to figure 3 and we have changed the title to “mRNA Expression Levels (Log of fold change)”and also the figure 4 to “microRNA Expression Levels (Log of fold change)”
9-Manuscript is well written; however, minor language improvement is needed throughout the manuscript.
Answer: As requested, we have carefully revised the manuscript and made several minor language improvements.

Round 2
Reviewer 2 Report
The authors have provided the appropriate explanation and improvements as suggested.